# Surface Properties of Ti65Zr Alloy Modified with TiZr Oxide and Hydroxyapatite

**DOI:** 10.3390/nano14010015

**Published:** 2023-12-20

**Authors:** Elinor Zadkani Nahum, Alex Lugovskoy, Svetlana Lugovskoy, Alexander Sobolev

**Affiliations:** Department of Chemical Engineering, Ariel University, Ariel 4070000, Israel; elinorna@ariel.ac.il (E.Z.N.); lugovsa@ariel.ac.il (A.L.)

**Keywords:** anodizing, nanotubes, growth kinetics, corrosion resistance, hydroxyapatite, biomedical applications

## Abstract

Titanium-zirconium dioxide nanostructures loaded by hydroxyapatite were produced on the surface of Ti65Zr alloy. The alloy was treated by anodization with the subsequent immersion in calcium glycerophosphate (CG) solutions. The resulting surfaces present TiO_2_-ZrO_2_ nanotubular (TiZr-NT) structures enriched with hydroxyapatite (HAP). The nanotube texture is expected to enhance the surface’s corrosion resistance and promote integration with bone tissue in dental implants. The TiZr-NT structure had a diameter of 73 ± 2.2 nm and a length of 10.1 ± 0.5 μm. The most favorable result for the growth of HAP in Hanks’ balanced salt solution (Hanks’ BSS) was obtained at a CG concentration of 0.5 g/L. Samples soaked in CG at a concentration of 0.5 g/L demonstrated in a decrease of the contact angles to 25.2°; after 3 days of exposure to Hanks’ BSS, the contact angles further reduced to 18.5°. The corrosion studies also showed that the TiZr-NT structure soaked in the CG = 0.5 g/L solution exhibited the best corrosion stability.

## 1. Introduction

Titanium and its alloys are the most common [1] and frequently chosen material for the manufacture of dental and orthopedic implants [2,3], as this was also confirmed by statistical data published by the National Center for Biotechnology Information (NSBI). However, commercial pure titanium metal (CP-Ti) has relatively low fatigue properties [4], which often leads to the destruction of the implanted material in the oral cavity [5]. Hence, to enhance the mechanical characteristics, alloying agents were incorporated into the metal composition. This facilitated an improvement in mechanical properties while preserving the potential for cell proliferation in the formation of bone tissue.

Zirconium-containing titanium alloys have demonstrated attractive properties to be used in orthopedic and dental surgery [6,7]. Titanium and zirconium possess exceptional corrosion resistance due to forming a natural oxide film, allowing easier integration with the bone matrix. These alloys also have a fatigue strength that is 13–42% better than CP-Ti [8].

With all the listed advantages of Ti alloys, there is a significant drawback, namely their bio-inertness, which means that the integration of the metal with living tissues is poor. Therefore, creating a bioactive surface layer consisting of TiO_2_ and HAP is one of the most promising methods for improving surface properties. Various surface modification methods, such as plasma spray, sol-gel, plasma electrolytic oxidation, and others, have been proposed [9,10,11,12,13,14]. These methods allow the application of a bioactive coating while significantly improving its hydrophilicity and corrosion resistance [15]. An interesting approach to the production of a layer of TiZr-NT on the implant’s surface is electrochemical anodization [16]. Typically, pure titanium is used as a base metal for obtaining ordered nanotubular (NT) structures, and initially, an amorphous nanotube framework form. The typically bioinert nature of TiO_2_ stems from their amorphous structure, necessitating an additional processing step to attain a crystalline form. To transform the amorphous structure of TiO_2_ into a crystalline structure [17], heat treatment at 450 °C is usually used. This anatase structure enhances the surface’s bioactive characteristics by accelerating bone structure growth rate. In contrast to Cp-Ti and many other alloys, anodizing TiZr alloys does not require post-processing, since crystalline mixtures of TiO_2_ and ZrO_2_ are formed immediately during the oxidized layer formation. The electrochemical method is based on a scalable and cost-effective process for coating medical implants with complex geometric shapes and sizes (including screws, plates, nails, and wires) [18]. Layers of synthesized nanotubes are chemically inert and have an acceptable mechanical strength, large surface area, and excellent load-bearing capacity. In addition, the nanotubular structure of the anodic oxide is an ideal matrix for doping with various antibacterial, medicinal, and bioactive components [19,20]. For these reasons, implants with nanotubular structures have attracted enormous attention in recent years [21,22].

Loading HAP into NT structures is usually a complex and multi-step procedure that requires careful control and adherence to technological parameters at each process stage. A. Kodama et al. [23] employed a multi-step procedure involving soaking and washing the NT configuration of TiO_2_. Diverse strategies for loading HAP involve hybrid techniques [24], such as the photocatalytic deposition of HAP, treatment with Ca and P, coupled with their degradation, and subsequent formation of HAP. Electrochemical processing methods [25,26] are extensively employed for filling the NT structure, utilizing electrolytes containing Ca and P. Implementing these methods involves intricate technological processes and necessitates strict adherence to all specified parameters.

Despite the technological process’s complexity, using these methods does not allow one to control the level of filling of the TiZr-NT structure with HAP. This leads to a decrease in the total contact area, thereby reducing the osseointegration properties of the surface.

This study outlines an innovative approach for generating an oxide layer on a Ti65Zr alloy coated with TiZr-NT and loaded with HAP. The resultant NT structure consists of a blend of TiO_2_ and ZrO_2_, and the HAP loading method is technologically straightforward. It involves immersing the NT structure in a single-component electrolyte, allowing for the filling of the NT structure. The proposed strategy offers the potential to enhance the hydrophilic and osseointegration properties of the implanted material’s surface through the one-step formation of TiO_2_ and ZrO_2_ crystalline structures, along with the managed introduction of HAP into the NT structure. The study includes examining structural characteristics (XRD, SEM), wettability, corrosion resistance, and the rate of HAP formation in Hanks’ BSS.

## 2. Materials and Methods

### 2.1. Experimental Procedures

Square-shaped 10 × 10 × 1 mm specimens of the Ti65Zr alloy were obtained from American Elements. Before experimentation, the samples were cleaned, degreased with acetone, and dried on air. After this, the samples’ surfaces underwent electropolishing [27] using a mixture (9:1) of concentrated acetic acid (CH_3_COOH) and perchloric acid (HClO₄) (Rhenium, Modi’in-Maccabim-Re’ut, Israel). Polishing was performed using a pulsed power supply (MP2-AS 35, Magpulls, Sinzheim, Germany) with a pulse length of 10 s at 40 V and 60 V. The total process time duration was 5 min at a temperature of 25 °C, with continuous stirring at 300 rpm. Titanium was a counter electrode (99.99%, Holland Moran, Yehud-Monosson, Israel).

The anodizing procedure used the same setup and settings as before. The anodizing electrolyte was prepared from ethylene glycol (C_2_H_6_O_2_) with 0.3 wt.% ammonium fluoride (NH_4_F), 0.66 wt.% sodium acetate (CH_3_COONa), and 2 wt.% distilled water (Rhenium, Israel). An electrolyte solution was made by mixing the required amount of distilled water with NH_4_F. This combination was then mixed with CH_3_COONa, which had previously been dissolved in ethylene glycol (EG). The electrolyte was stirred at 300 rpm on a magnetic stirrer for an hour to ensure a homogeneous composition of the electrolyte. Anodization occurred over 2 h at 60 V. Following the treatment, the samples were washed with distilled water and dried.

Post-anodic treatment, the samples were immersed in 0.5–5 g/L CG solutions with constant stirring of 300 rpm for 24 h. After that, each test sample was immersed in 20 mL Hanks’ BSS at 37 ± 0.5 °C for 3 days. To maintain a consistent ion concentration throughout the study, the solution was replaced on the second day with a fresh portion, discarding the ‘old’ solution.

### 2.2. Analytical Methods

To assess the morphology and elemental composition of the surface of the samples, SEM (FEI Magellan™ 400 L, Thermo Fisher Scientific Inc., Waltham, MA, USA) with an energy dispersive spectroscopic analyzer EDS (Aztec Oxford Instruments, Concord, MA, USA) was used. Previously, a nano-sized layer of chromium was applied to the surface of the samples under study using a turbomolecular pumped coater Quorum Q150T (Quorum Technologies Ltd., Laughton, UK) providing a lower vacuum down to 5 × 10^−5^ mbar and current 100 mA for 10 s. The applied chromium layer does not affect the surface morphology but enhances its conductive properties thus preventing it from electrostatic charging during SEM analysis. Surface morphology examinations were conducted using a through the lens detector (TLD) at 100–500 k× magnifications and an accelerating voltage of 5 kV.

For phase analysis of the prepared samples, an XRD-SmartLab SE (Rigaku, Tokyo, Japan) equipped with a Cu-Kα gun with a wavelength (λ) of 1.54059 Å at a current of 30 mA and 40 kV was used. The scanning parameters were set at 0.03°/step with a scanning speed of 0.3°/min.

To determine the surface wettability (CA) 6 μL drops of Hanks’ BSS, distilled water, and glycerol were applied to their surface. The surface energy (*γ*) of the coatings was calculated using the Owens–Wendt method [28], which assumes that the surface free energy *γ_S_* is the sum of the dispersion component *γ_S_^D^* and the polar component *γ_S_^P^*_._
(1)$$\gamma_{l}\left(1+\cos\theta \right)=2\sqrt{\gamma_{l}^{D}}\gamma_{s}^{D}+2\sqrt{\gamma_{l}^{P}}\gamma_{s}^{P}$$

To calculate the surface free energy, the measured contact angle (*θ*) was used, as well as the surface tension of the liquids ($\gamma_{l}$), which is the sum of the dispersion ($\gamma_{l}^{D}$) and polar ($\gamma_{l}^{P}$) components of the surface tension. The surface tension of distilled water ($\gamma_{l}$ = 72.8 mJ∙m^−2^) and glycerol ($\gamma_{l}$ = 63.4 mJ∙m^−2^) is known, as are their dispersion (21.8 mJ∙m^−2^; 37 mJ∙m^−2^) and polar (51 mJ∙m^−2^; 26.4 mJ∙m^−2^) components [29].

Experiments were performed at room temperature, and results were based on an average of five measurements.

Potentiodynamic polarization curves (PPC) and electrochemical impedance spectroscopy (EIS) were carried out using a PARSTAT 4000A (Princeton Applied Research, Oak Ridge, TN, USA) equipped VersaStudio V. 2.65.2 software. Experiments were carried out at 37 ± 0.5 °C in Hanks’ BSS. Electrochemical studies were carried out using a three-electrode cell. The reference electrode was Ag/AgCl_sat_, and the counter electrode was a platinum sheet. To stabilize the open circuit potential (OCP), samples were immersed in Hanks’ BSS for an hour before testing.

During the experiment, PPC was recorded at 1 mV/s in a potential range of ±500 mV relative to the OCP. Based on experimental data of corrosion potentials (E_corr_) and currents (I_corr_), as well as anodic and cathodic slopes of the linear sections (β_a_ and β_c_), using the Stern–Geary equation [30], the polarization resistance (Rp) was calculated (2) for each studied sample.
(2)$$Rp=\frac{\beta_{a}\cdot \beta_{c}}{2.3\cdot I_{corr}\cdot (\beta_{a}+\beta_{c})}$$

EIS data scans were conducted at 100 kHz to 0.1 Hz vs. OCP with a root mean square (RMS) amplitude of 5 mV. For analysis of EIS results, EC-Lab^®^ V. 11.10 software was used. The codes assigned to the examined specimens are detailed in Table 1.

To check the reproducibility of the results and determine the average value and standard deviation, four identical samples, S1–S5, were prepared.

## 3. Results and Discussions

### 3.1. Anodizing Process

The experimental conditions employed for anodizing play a pivotal role in achieving well-organized arrays of TiZr-NT with specified pore sizes and wall thicknesses. The formation of these TiZr-NT arrays in a fluoride-based electrolyte involves the following processes: (a) the oxidation of TiZr alloy (reactions (3) and (4)) and (b) the chemical dissolution of the formated titanium-zirconium dioxide with the creation of a tubular structure (reactions (5) and (6)). The critical aspect in initiating and developing TiZr-NT is the balance between the rates of these two competing processes (a) and (b). In a broader context, the anodization process presents the following reactions [31]:Me → Me^4+^ + 4e^−^
(3)
Me^4+^ + 2H_2_O → MeO_2_ + 4H^+^
(4)
Me^4+^ + 6F^−^ → [MeF_6_]^2−^
(5)
MeO_2_ + 6F^−^ + 4H^+^ → [MeF_6_]^2−^ + 2H_2_O(6)
where: Me = Ti or Zr.

Figure 1 shows the current plot recorded during anodization at 60 V. Theoretically, the following processes can simultaneously occur on the metal surface: formation of a barrier layer and TiZr-NT structure. The rapid drop in current during the initial period of anodization suggests that the main process is forming a barrier layer due to reactions (3) and (4). Upon transition to the quasi-stationary mode (7.8 mA), the formation of the barrier layer ends, and TiZr-NT begins according to reactions (5) and (6).

The current plot exhibits a characteristic pattern associated with electrolytes relying on EG. The significant EG viscosity notably affects reagent diffusion into the reaction area [32], leading to a slowdown in the TiO_2_ and ZrO_2_ dense layer’s chemical and electrochemical dissolution processes.

### 3.2. Morphology

Figure 2 illustrates typical SEM images captured from Ti65Zr samples after anodization at a potential of 60 V.

The samples obtained post-anodizing showed a TiZr-NT structure with a pore diameter of 73 ± 2.2 nm, wall thickness of 23 ± 1.1 nm, and length of 10.1 ± 0.5 µm. Chemical analysis of the surface using EDS on S1 (refer to Table 2, Spectrum 1 (Sp. 1)) indicated an almost stoichiometric presence of Ti, Zr, and O.

Figure 3 presents the morphology and cross-section of sample S2 after exposure to CG solutions with a concentration of 0.5 g/L.

The EDS method’s point elemental analysis and the surface morphology examination of S2 revealed that the outer part of the pore (Sp. 2 and Sp. 3) contains Ti, Zr, and O. In contrast, the interior of the pores of sample S2 (Sp. 4 in Figure 3 and Table 2) comprises a blend of Ti, Zr, O, Ca, and P. Notably, the ratio of Ca to P inside the porous structure of the NT structure of titanium-zirconium falls in the range of 1.67–1.74 (as indicated in Table 2), closely matching the chemical composition of HAP (Ca/P = 1.67) [33].

Figure 4 depicts the surface morphology and cross section of sample S3 after soaking it in CG = 5 g/L solution.

A closer look at the sample’s surface (Figure 4) reveals crystallized HAP (Table 2, Sp. 5), covering parts of the nanostructured surface, which should favor more efficient cell proliferation and bone tissue growth [34]. The cross-section of the samples shows a different morphology than for S2, namely pores filled with HAP (Table 2, Sp. 7). The chemical composition of the pore walls is similar for S1 and S2 and consists of zirconium and titanium oxides (Table 2, Sp. 6).

The hydrolysis of CG and the formation of HAP on the walls of TiZr-NT (Figure 3) and inside them (Figure 4) proceeds through the following reactions:C_3_H_7_CaO_6_P + H_2_O → CaHPO_4_ + C_3_H_5_(OH)_3_(7)
7CaHPO_4_ + H_2_O → Ca_5_(PO_4_)_3_OH + 2Ca(H_2_PO_4_)_2_(8)

The hydrolysis of CG proceeds through two consecutive reactions labeled as reactions (7) and (8). In the initial reaction (7), dicalcium phosphate (CaHPO_4_) and glycerol (C_3_H_5_(OH)_3_) are formed. Subsequently, in the second reaction (8), CaHPO_4_ undergoes hydrolysis to produce HAP and monocalcium phosphate (Ca(H_2_PO_4_)_2_). Due to its insolubility, HAP precipitates inside the TiZr-NT during the gradual hydrolysis process. Ca(H_2_PO_4_)_2_ is highly soluble in water and physiological liquids.

The thickness of the formed HAP on the pore walls of sample S2 is measured at 23 ± 1.1 nm and presents a nano-porous structure. However, with an increase in the concentration of CG to 5 g/L (S3), the pores become filled with HAP during soaking. Coated samples S4 and S5 were exposed to Hanks’ BSS for 3 days at 37 ± 0.5 °C to evaluate the bioactivity indirectly. The results of the structure changes are shown in Figure 5 and Figure 6.

S4 experienced the formation of a HAP (Figure 5a) after 3 days of exposure to Hanks’ BSS, which was also confirmed by EDS analysis (Sp. 8, Table 2) and XRD (Figure 6d). The rapid formation of the HAP structure on the coated surface may indicate high biological activity and good osseointegration potential. With similar holding parameters for S5, the rate of HAP formation was much lower than confirmed by the surface morphology (Figure 5b) and EDS analysis (Sp. 9, Table 2). Therefore, the most preferable is the formation of HAP on the walls of pores as centers of nucleation and growth of bone tissue.

### 3.3. XRD Analysis

XRD results presents in Figure 6 for Ti65Zr, S1–S5 samples.

The Ti65Zr initially exhibited characteristic phases of a titanium-zirconium alloy (ICDD 01-072-3352). Following the anodizing process (as shown in Figure 6b), an amorphous halo in the range of 20°–38° and peaks of crystalline Ti_2_ZrO (ICDD 01-075-1739) emerged. The observed amorphous structure in the titanium-zirconium oxide is likely attributed to the porous layer, while the barrier layer consists of a crystalline structure similar to titanium nanotubular layers [35].

After S2 is immersed in the CG solution, HAP (ICDD 00-064-0738) is formed on the surface of the TiZr-NT structure, as shown in Figure 3. This is confirmed by the X-ray diffraction pattern shown in Figure 6c with characteristic 2θ angles of 25.8°, 31.7° and 53.4°. The emergence of HAP aligns with the earlier chemical EDS analysis (Table 2, Sp. 2–Sp. 4).

After exposure to the sample S4 in Hanks’ BSS, the phase composition remains relatively stable (Figure 6d) and consists of an amorphous-crystalline HAP layer, confirmed by the morphology and surface EDS analysis in Figure 5a. Similar results were obtained for samples S3 and S5.

### 3.4. Contact Angle

A key role in successfully integrating implants with bone tissue is achieved by achieving the required initial stability of the implanted material [36]. This parameter is often achieved by changing implants’ geometric shapes and sizes [37]. The stabilized implant begins to actively interact with the body’s biological fluids, which are responsible for forming a new bone structure. The rate of formation and quality of adhesion of mononuclear cells and their further transformation into macrophages [38] is very closely related to the structural features of the surface, as well as its wettability and surface energy (SE). To study surface wettability, a series of contact angle (CA) measurements were carried out using Hanks’ BSS at 37 ± 0.5 °C.

CA measurements present in Figure 7 and Table 3.

The Ti65Zr sample had the highest CA = 84.1 ± 1.3° of all those studied (Table 3, Figure 7a) and the lowest SE γ = 41.6 ± 1.2 mJ∙m^−2^. The anodized S1 slightly reduced the CA = 70.2 ± 1.2°. On sample S2, after exposure to a CG = 0.5 g/L solution, a nanoporous HAP structure was formed on the bottom and walls of TiZr-NT, significantly reducing the CA = 25.2 ± 1.3°. With an increase in the concentration of CG = 5 g/L, the CA increased slightly to 29.3 ± 1.0° (sample S3). The deterioration of the hydrophilic properties of the surface is associated with the complete filling of the TiZr-NT structure (Figure 4).

When the samples were kept in Hanks’ BSS, a uniform layer of HAP was formed on the S4 (Figure 5a) with CA = 18.5 ± 0.5° and maximum SE = 74.8 ± 0.6 mJ∙m^−2^. In the case of sample S5 (Figure 5b), the formed coating has a non-uniform structure and higher CA = 24.3 ± 0.8°.

Based on the results presented in Table 3, the SE varies inversely with the CA. The increase in surface energy and decrease in CA is caused by liquid penetration, which is facilitated by capillary forces [39]. Accelerating the penetration of biological fluid into the porous structure promotes the formation of a bone matrix and improves the osseointegration [40] properties of the surface.

### 3.5. Electrochemical Test

Electrochemical studies were carried out using PPC and EIS tests in Hanks’ BSS at 37 ± 0.5 °C. The PPC results are shown in Figure 8 and summarized in Table 4.

As shown in Figure 8, the corrosion current (I_corr_) and potential (E_corr_) values were obtained by approximating the linear sections of the cathode and anodic curves. The anodic (β_a_) and cathodic (β_c_) slopes of the linear sections of the curves in semilogarithmic coordinates are associated with the activation energy of the anodic or cathodic reaction and their occurrence rates. Based on the approximation of experimental data (Figure 8, Table 4) the polarization resistance was calculated using the Stern–Geary equation.

Based on the data obtained, the Ti65Zr sample exhibited the most significant tendency to corrosion, namely a high corrosion current (I_corr_ = 386 ± 3 nA∙cm^−2^) and low polarization resistance (Rp = 296.9 kOhm∙cm^−2^). Anodized sample S1, due to the formation of a two-layer oxide coating, significantly reduced I_corr_ while increasing its corrosion resistance by almost two times. A further increase (Samples S2, S3) in polarization resistance and a decrease I_corr_ is associated with the filling of TiZr-NT with HAP and the formation of an additional layer (Figure 3 and Figure 4), increasing corrosion protection efficiency.

To be able to monitor the dynamics of the corrosion behavior of the formed coatings, samples S4 and S5 were exposed for 3 days in Hanks’ BSS. The results obtained show a decrease in I_corr_ and an increase in polarization resistance, which is a consequence of the formation of an additional HAP layer (Figure 5).

In general, samples S1–S5 demonstrate a tendency to increase polarization resistance in the following order: S1 = 531.5 kOhm∙cm^−2^; S2 = 549.2 kOhm∙cm^−2^; S3 = 551.6 kOhm∙cm^−2^; S4 = 637.1 kOhm∙cm^−2^; S5 = 586.1 kOhm∙cm^−2^.

The EIS of S1–S5 is presented in Figure 9, and their equivalent electrical circuit (EEC) is shown in Figure 10.

Based on the results of studying the morphology of the coating (Figure 2, Figure 3, Figure 4 and Figure 5) of samples S1–S5 and the dependence of the phase angle on frequency (Figure 9c), it was established that the coating has a two-layer structure. To describe the obtained spectra, a double chain of components R-CPE was used (Figure 10), where the element R_2_-CPE_2_ [41] characterizes the barrier non-porous layer of the coating (low-frequency region, Figure 9c), and R_1_-CPE_1_ characterizes the outer nanotubular layer (mid-frequency region, Figure 9c). The electrolyte resistance was designated R_s_ and was 37 ± 2 Ohms.

**Figure 10 nanomaterials-14-00015-f010:**
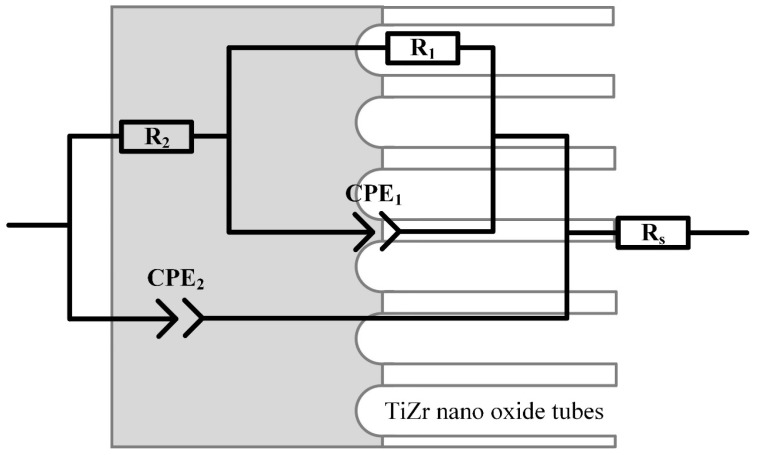
Equivalent electrical circuit for approximation of experimental Electrochemical Impedance Spectroscopy spectra [42].

A constant phase element CPE was used to model and describe the capacitive properties of the coating [43,44]. The values of the EEC fitted to the measured EIS spectra are provided in Table 5.

In Figure 9a, the presence of depressed semicircles aligns with the existence of two capacitive loops with similar time constants. The phase angles in Figure 9c, measuring 75° for the inner layer and 45° for the outer layer, further highlight diffusion retardation in the nanotubes and increased activation control at the inner layer.

The anodized samples S1 have the lowest porous and barrier layer resistance with high capacitive properties of the coating in the series of samples S1–S5. In the process of filling samples S2 and S3 with HAP (according to reactions (7) and (8)), there is a slight increase in the resistance of the porous layer and a decrease in its capacitive characteristics. This phenomenon is associated with different types of filling of the TiZr-NT structure and the interaction of HAP with the surface of the nanotubes. The resistances of the barrier layer for samples S2 and S3 increased slightly compared to sample S1, which may indicate the encapsulation of defects by a layer of HAP.

The dimensionless coefficients n_1_ and n_2_ for S1–S3 have close values in the range of 0.87–0.96 and exhibit obvious capacitive properties. The found impedance modules for S1–S3 have somewhat relative values and correlate with Rp obtained by the PPC method. On S4 and S5 (Figure 9c), there is a slight shift in the spectrum to the low-frequency region, which indicates a change in the morphology of the surface layer. Based on its structural analysis (Figure 5), the change in morphology occurs due to the formation of a HAP layer on the surface of the samples. Capacitive characteristics of the TiZr-NT oxide coating decrease after 3 days, indicating a slowdown in corrosion processes. Sample S4 shows a significant increase in impedance modulus (Table 5), suggesting the formation of a more uniform surface layer after 3 days of exposure. In summary, the TiZr-NT oxide layer loaded with HAP will likely support osseointegration by aiding in the nucleation and growth of HAP.

## 4. Conclusions

In this study, a meticulously structured TiZr-NT oxide layer, composed of TiO_2_ and ZrO_2,_ was generated on a Ti65Zr specimen with a one-step electrochemical anodizing. The resulting TiZr-NT structures exhibited a diameter of 73 ± 2.2 nm and a length of 10.1 ± 0.5 µm. The TiZr-NT oxide layer was enriched with HAP by immersing the samples in CG with various concentrations. It was followed by a stepwise hydrolysis process to enhance bioactivity and its structural features.

SEM analysis of samples treated with a CG = 0.5 g/L showed that the formation of a nanoporous structure of HAP is observed on the walls of TiZr-NT and not in bulk as with CG = 5 g/L. The nanoporous structure of HAP distributed along the pore walls gives the smallest CA before exposure to Hanks’ BSS (CA = 25.2° ± 1.3°) and after (CA = 18.5° ± 0.5°). The applied method of modifying the surface of samples in CG = 0.5 g/L significantly increased the rate of formation of HAP and improved the corrosion properties Z_f→0 Hz_ = 637.9 kΩ∙cm^2^.

The formation of a nanoporous structure on the walls of TiZr-NT occurs as a result of the following reactions:C_3_H_7_CaO_6_P + H_2_O → CaHPO_4_ + C_3_H_5_(OH)_3_
7CaHPO_4_ + H_2_O → Ca_5_(PO_4_)_3_OH + 2Ca(H_2_PO_4_)_2_

Loaded HAP as crystallization centers in the TiZr-NT structure improves the bioactive properties by accelerating the osseointegration process.

## Figures and Tables

**Figure 1 nanomaterials-14-00015-f001:**
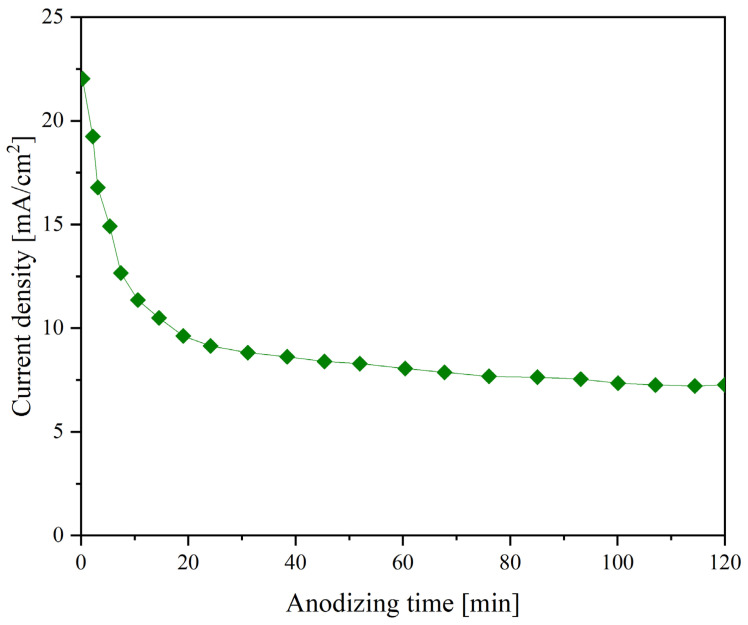
Current density plot during anodizing process at 60 V.

**Figure 2 nanomaterials-14-00015-f002:**
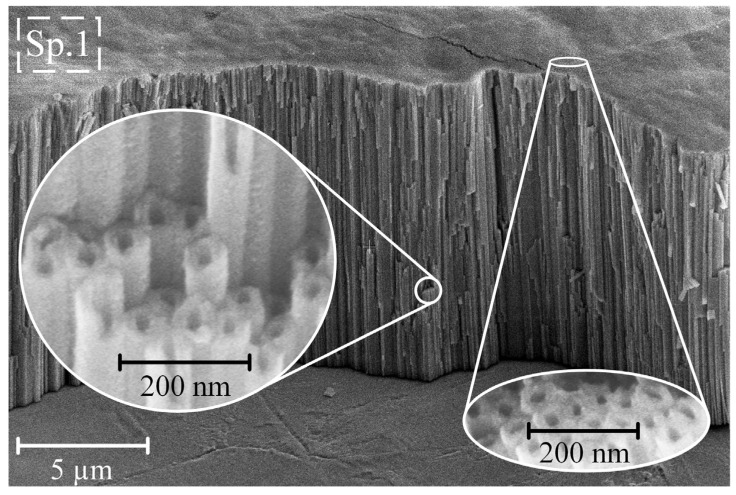
Typical coating morphology and cross-section of anodized sample S1 (60 V).

**Figure 3 nanomaterials-14-00015-f003:**
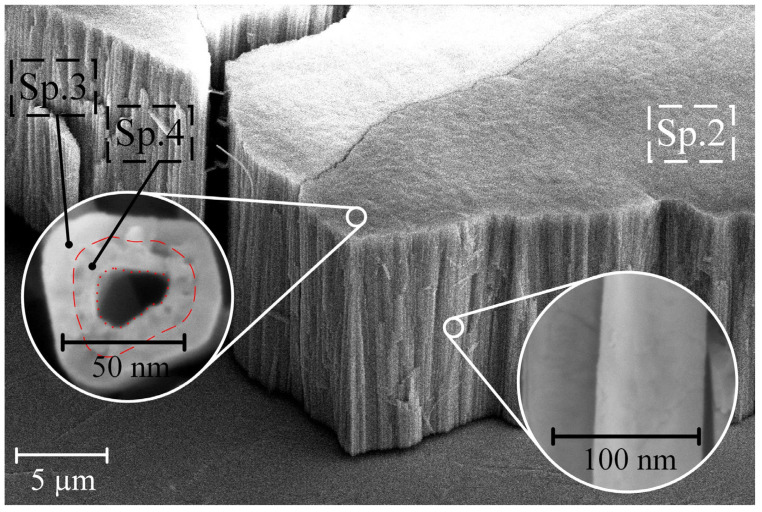
Morphology cross-section of samples S2 after soaking in CG = 0.5 g/L solution.

**Figure 4 nanomaterials-14-00015-f004:**
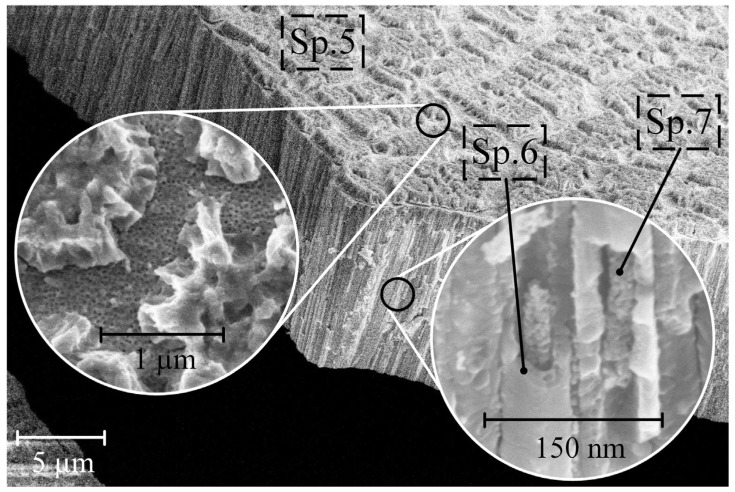
Surface morphology and cross-section of samples S3 after soaking in CG = 5 g/L solution.

**Figure 5 nanomaterials-14-00015-f005:**
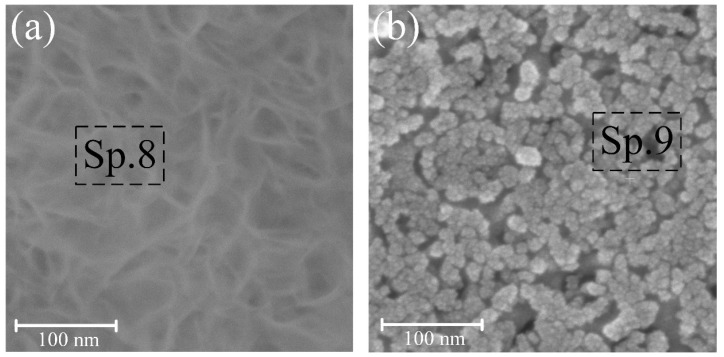
Surface morphology of samples pre-treated in 0.5 g/L (**a**) and 5 g/L (**b**) CG solution after 3 days of exposure at 37 ± 0.5 °C in Hanks’ balanced salt solution.

**Figure 6 nanomaterials-14-00015-f006:**
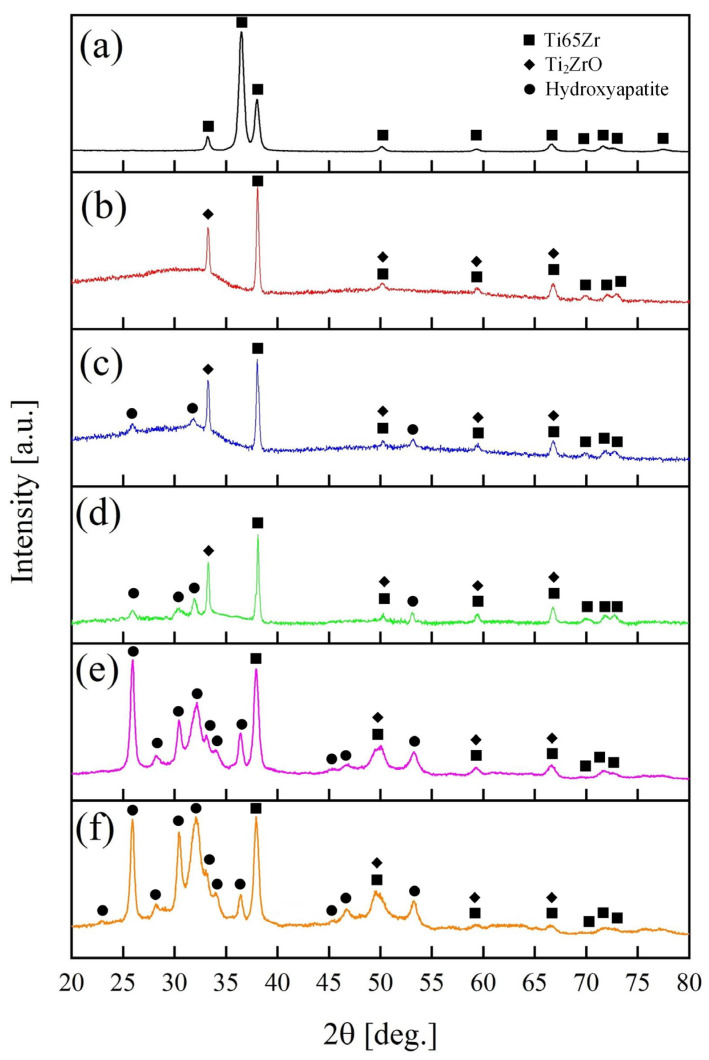
XRD patterns of (**a**) untreated Ti65Zr; (**b**) S1; (**c**) S2; (**d**) S3; (**e**) S4; (**f**) S5.

**Figure 7 nanomaterials-14-00015-f007:**
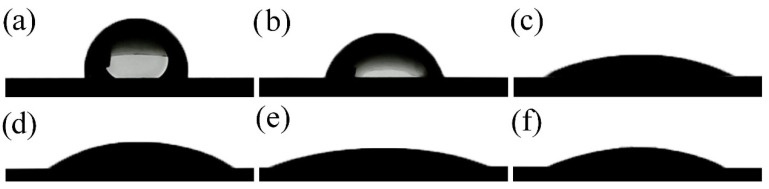
Drops of Hanks’ balanced salt solution on the surface of (**a**) Ti65Zr; (**b**) S1; (**c**) S2; (**d**) S3; (**e**) S4; (**f**) S5.

**Figure 8 nanomaterials-14-00015-f008:**
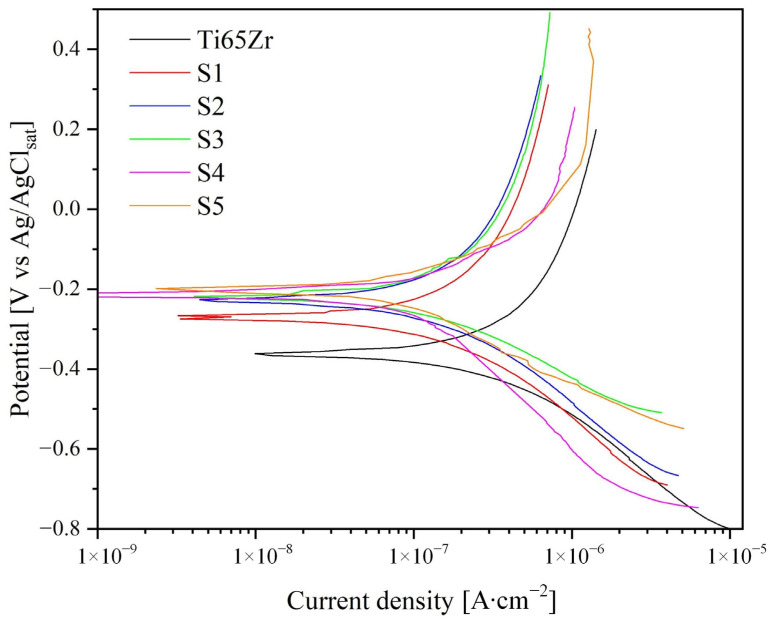
Potentiodynamic polarization curves for Ti65Zr and S1–S3.

**Figure 9 nanomaterials-14-00015-f009:**
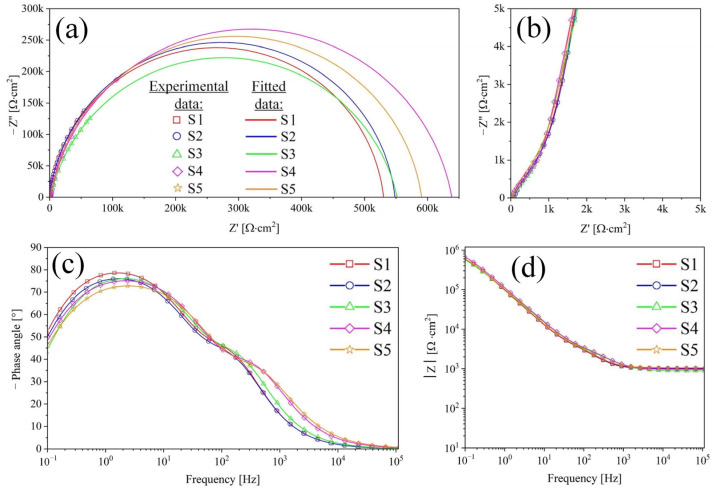
Nyquist (**a**,**b**) and Bode (**c**,**d**) plots for samples S1–S5.

**Table 1 nanomaterials-14-00015-t001:** Sample codes with corresponding processing.

	Anodizing	Soaking in Calcium Glycerophosphate Solution		Exposure Test in Hanks’ Balanced Salt Solution	
		C = 0.5 [g/L] 24 h	C = 5 [g/L] 24 h	0 Days	3 Days
Ti65Zr	Without treatment				
S1	+				
S2	+	+		+	
S3	+		+	+	
S4	+	+			+
S5	+		+		+

**Table 2 nanomaterials-14-00015-t002:** Point EDS analysis for S1–S5 samples.

	Sample Number	Ti [at.%]	Zr [at.%]	O [at.%]	P [at.%]	Ca [at.%]	Ca/P
Sp. 1	S1	22.3	23.5	54.2	-	-	-
Sp. 2	S2	22.1	23.3	54.2	0.15	0.25	1.60
Sp. 3	S2	22.5	22.8	54.4	0.1	0.20	2.00
Sp. 4	S2	19.8	21.8	54.2	1.70	2.50	1.67
Sp. 5	S3	17.0	17.9	50.6	6.30	8.20	1.30
Sp. 6	S3	22.4	23.3	54.3	0.23	0.4	1.72
Sp. 7	S3	19.1	20.1	52.1	3.18	5.47	1.72
Sp. 8	S4	-	-	65.8	12.8	21.4	1.67
Sp. 9	S5	8.2	8.6	68.2	6.2	8.8	1.42

**Table 3 nanomaterials-14-00015-t003:** SE (γ) and CA values for Ti65Zr, S1–S5 samples.

	Contact Angle [°]			SE–γ
	Hanks’ BSS	Water	Glycerol	[mJ∙m^−2^]
Ti65Zr	84.1 ± 1.3	76.3 ± 1.1	90.3 ± 1.2	41.6 ± 1.2
S1	70.2 ± 1.2	63.1 ± 1.1	77.4 ± 1.3	51.9 ± 1.2
S2	25.2 ± 1.3	16.8 ± 1.4	33.2 ± 1.2	74.5 ± 1.3
S3	29.3 ± 1.0	21.4 ± 0.9	37.6 ± 1.0	74.1 ± 1.1
S4	18.5 ± 0.5	10.3 ± 0.7	26.1 ± 0.9	74.8 ± 0.6
S5	24.3 ± 0.8	16.4 ± 1.0	32.6 ± 1.1	74.7 ± 1.3

**Table 4 nanomaterials-14-00015-t004:** Electrochemical parameters of Ti65Zr, S1–S5 samples studied by the Potentiodynamic polarization method.

	E_corr_ vs. Ag/AgCl_sat_ [mV]	I_corr_ [nA∙cm^−2^]	β_a_ [mV·dec^−1^]	−β_c_ [mV·dec^−1^]	R_p_ [kΩ·cm^−2^]
Ti65Zr	−363 ± 5	386 ± 3	705	421	296.9
S1	−269 ± 6	171 ± 4	679	302	531.5
S2	−225 ± 3	159 ± 2	441	336	549.2
S3	−220 ± 3	150 ± 2	352	223	551.6
S4	−219 ± 4	105 ± 3	259	379	637.1
S5	−213 ± 3	161 ± 2	189	178	586.1

**Table 5 nanomaterials-14-00015-t005:** Calculated parameters of ECC elements for S1–S5 samples.

Impedance Spectroscopy Parameters	S1	S2	S3	S4	S5
CPE_1_ × 10^−5^ [S·cm^−2^·s^−n^]	0.21	0.19	0.15	0.132	0.128
n_1_	0.94	0.93	0.87	0.89	0.92
R_1_ [kΩ·cm^2^]	1099	1140	1893	2097	3348
CPE_2_ × 10^−5^ [S·cm^−2^·s^−n^]	0.30	0.27	0.22	0.206	0.171
n_2_	0.96	0.94	0.86	0.95	0.903
R_2_ [kΩ·cm^2^]	529.0	547.1	549.4	635.8	586.7
Z_f→0 Hz_ [kΩ·cm^2^]	530.1	548.2	551.3	637.9	589.9
χ^2^ × 10^−4^	1.2	2.4	2.5	3.1	3.4

## Data Availability

All the data supporting the findings of this study are available within the article.

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
