# Peer review of "Surface Properties of Ti65Zr Alloy Modified with TiZr Oxide and Hydroxyapatite"

_nanomaterials, 2023, doi:10.3390/nano14010015_

Round 1

Reviewer 1 Report

Comments and Suggestions for Authors

Comments to authors

The manuscript entitled "Surface properties of Ti65Zr alloy modified with TiZr oxide and hydroxyapatite” deals with surface properties of the anodized TiO2-ZrO2 nanotubes (TiZr-NT) deposited on Ti65Zr after immersing in calcium glycerophosphate (CG), which results in hydroxyapatite (HAP) growth. The modified surface is studied regarding its corrosion resistance and bioactive properties such as HAP growth rate and osseointegration process.

Although, in literature there are many methods of loading HAP into TiO2 nanotubes structures, these methods do not allow one to control the level of nanotubes filling with HAP. This work overcomes this drawback through the generation of a blend of nanotubular Ti and Zr oxide layer along with the managed introduction of HAP into nanotubular structure by immersing the oxide layer in a single component electrolyte (CG). This is the novelty and manuscript’s strength point.

However, there is a point that authors have to take under consideration:

The figures 2 and 3, show that the anodized TiZr-NT layer is not fully integrated or non-adhesive with the underlying Ti65Zr layer. So, it can peel or chip away. The authors have to explain, does it happen intentionally or accidentally?   

Minor revisions:

11. In table 2 after the 1st column, a column with samples S1-S5 must be        added in order to show the correspondence of EDS points spectra with the sample’s number.

22. The caption “Figure 2” in page 5 must change to “Figure 1”             

Reviewer 2 Report

Comments and Suggestions for Authors

The paper is well written and presents good results. Figures are good. Hypothesis and implementation is good. However, the repeatability of the results are not presented in most of the figures. This is the only concern with this paper. If the authors can show that the experiments are repeatable in each figure then it can be accepted for publication.

Reviewer 3 Report

Comments and Suggestions for Authors

In this manuscript, “Surface properties of Ti65Zr alloy modified with TiZr oxide and hydroxyapatite” by Nahum et al. reports the fabrication and characterization of an oxide layer on a Ti65Zr alloy coated with TiZr-NT and loaded with HAP. A revision is required for this manuscript. Here are the comments and suggestions:

1.        The abstract should be revised.

2.        Several paragraphs contain only single sentence.

3.        Sample number can be added to the Table 2, and the results of sp.8 seems disagreed with that of Fig. 6.

4.        Please add results of sample S5 to Fig. 6.

Reviewer 4 Report

Comments and Suggestions for Authors

The manuscript entitled: “Surface properties of Ti65Zr alloy modified with TiZr oxide 2

and hydroxyapatite”, reference:

It is an interesting manuscript, easy to follow. Nevertheless, in my opinion its clarity and impact could be considerably improved. Please allow me to provide some suggestions:

1.      The initial introduction should comprise some recent statistics of the use of titanium as dental and orthopedic implants, particularly due to the age of the used references to support their widespread use (over than ten and twenty years old). Furthermore, titanium implant issues should also possess recent statistics and more importantly, should have relevant references to support it (line 30 to 35).

2.      The Material and Methods in my opinion it is too brief for clarity and their description should be in the same order as the presented results (or vice-versa). In addition, please correct me if I am wrong, but for instance:

a.      It is not clear the choice of using Hanks' BSS

b.      Line 117 and 118, the information for SEM EDS images is in my point of insufficient. In particular, there is no information of the treatment that the samples underwent prior to SEM-EDS, or the absence of treatment.

c.      The use of two different concentrations of CG

d.       

3.      Result sections issues:

a.      Figure 3 and Figure 4 insets do not possess scales. It should be as Figure 2.

b.      The authors should try to display a histogram with size distribution of the polyhedral structures for each of the 5 samples. The method of measuring and the discussion of the results should also be added.

c.       Why the XRD only comprises S1, S2 and S4?

d.      Why no statistical analysis were performed in Table 3?

e.      Why Figure 8 does not comprise samples S4 and S5?

f.       The parameters present in Table 4 are not adequately defined or explained, in my point of view.

g.      Equation 7 should be present in Materials and Methods section.

h.      Table 6 does not describe the parameters of the control and S1, why?

Additional suggestions:

Several acronyms and chemical formulas are not adequately defined.

Several units are not separated form their numerical value.

Line 90, please consider substituting the term “Rectangular” with “Rectangular prism”, since the described dimensions are 10×10×1 mm.

Line 196 to 197, no references used to support this statement.

Comments on the Quality of English Language

The English is correct, I could not find major flaws, nevertheless sometimes is to brief for clarity.

Round 2

Reviewer 2 Report

Comments and Suggestions for Authors

The authors have sufficiently amended the article based on the review comments. It can be accepted for publication.

Author Response

Thanks for your help and support.

Reviewer 3 Report

Comments and Suggestions for Authors

It seems more acceptable now.

Author Response

Thanks for your help and support.

Reviewer 4 Report

Comments and Suggestions for Authors

The manuscript was clearly improved. Nevertheless, I failed to denote an important issue: the information to determine the surface free energy is grievously insufficient, the authors must detail the equation used. More importantly which were the liquids used for this determination. I beg your pardon for the delay.

Please explain why the units kΩ âˆ™ cm2 is placed inside brackets within the text.

Answer provided by the authors: 3.a. Eventhough the authors do not present the gaussian distribution, I would like to recommend the addition of the number of measurements performed. Is this already present in the manuscript?
